# Near-Infrared Spectroscopy and Chemometrics for Effective Online Quality Monitoring and Process Control during Pelagic Fishmeal and Oil Processing

**DOI:** 10.3390/foods13081186

**Published:** 2024-04-13

**Authors:** María Gudjónsdóttir, Gudrún Svana Hilmarsdóttir, Ólafur Ögmundarson, Sigurjón Arason

**Affiliations:** 1Faculty of Food Science and Nutrition, University of Iceland, Nýi Garður, Sæmundargata 12, 102 Reykjavík, Iceland; gudrunsvana@hi.is (G.S.H.); olafuro@hi.is (Ó.Ö.); sigurjar@hi.is (S.A.); 2Matis Food and Biotech R&D, Vínlandsleid 12, 113 Reykjavík, Iceland

**Keywords:** near-infrared (NIR) spectroscopy, chemometrics, process monitoring, quality assessment, process optimization, product optimization

## Abstract

Near-infrared spectroscopy has become a common quality assessment tool for fishmeal products during the last two decades. However, to date it has not been used for active online quality monitoring during fishmeal processing. Our aim was to investigate whether NIR spectroscopy, in combination with multivariate chemometrics, could actively predict the changes in the main chemical quality parameters of pelagic fishmeal and oil during processing, with an emphasis on lipid quality changes. Results indicated that partial least square regression (PLSR) models from the NIR data effectively predicted proximate composition changes during processing (with coefficients of determination of an independent test set at $R_{CV}^{2}$ = 0.9938, RMSECV = 2.41 for water; $R_{CV}^{2}$ = 0.9773, RMSECV = 3.94 for lipids; and $R_{CV}^{2}$ = 0.9356, RMSECV = 5.58 for FFDM) and were successful in distinguishing between fatty acids according to their level of saturation (SFA ($R_{CV}^{2}=0.9928,\;RMSECV=0.24)$, MUFA ($R_{CV}^{2}=0.8291,\;RMSECV=1.49)$, PUFA ($R_{CV}^{2}=0.8588,\;RMSECV=2.11)$). This technique also allowed the prediction of phospholipids (PL $R_{CV}^{2}=0.8617,\;RMSECV=0.11$, and DHA ($R_{CV}^{2}=0.8785,\;RMSECV=0.89)\;$ and EPA content $R_{CV}^{2}=0.8689,\;RMSECV=0.62)$ throughout processing. NIR spectroscopy in combination with chemometrics is, thus, a powerful quality assessment tool that can be applied for active online quality monitoring and processing control during fishmeal and oil processing.

## 1. Introduction

Recently, pelagic fishmeal for use in fish and animal feed has been lowering in value due to the high competition with alternative protein meals. However, pelagic fishes, such as the Atlantic mackerel (*Scomber scombrus*), Atlantic herring (*Clupea harengus*), and blue whiting (*Micromesistius poutassou*), are nutritious raw materials containing several valuable fatty acids, essential proteins, and minerals [1]. This calls for increased utilization of pelagic fish raw materials and optimization of production processes of higher-value products, including products for human consumption. Traditional fishmeal and oil processes from pelagic fish species have remained unchanged for decades [1,2,3]. Moreover, the variation in water, protein, and lipid content in the processing streams during fishmeal processing of pelagic fishes is extreme due to the high variation in the raw material that enters the process [1,4,5]. The raw material often includes several species, cut-offs, heads, bones and guts, and the availability of species is highly seasonal [1,6]. The share of side-streams from pelagic fish processing to human consumption has increased in recent years and will increase significantly in the future. This expected demand of products from pelagic side-stream processing calls for effective raw material and process monitoring of the chemical changes occurring at each processing step. Such monitoring is critical for process optimization and fast decision making towards the production of higher-value products. However, traditional quality assessment methods performed in the laboratory are commonly sample destructive, time consuming, and expensive, which does not fit well into the need for fast decision making within the processing environment [4].

Near-infrared spectroscopy (NIR) is a fast, non-destructive analytical method that has been used for some time for the quality assessment of seafood, including the quality of the final fishmeal products [4,7]. The main chemical constituents of fish and fishmeal are rich in C-H, O-H, N-H, and S-H chemical bonds, which overtones and combination bands absorb strongly in the NIR spectral range [8]. NIR analysis has, therefore, been shown to be useful in predicting several chemical [4,7] and derived physical [9] and sensory quality parameters [10,11,12] in diverse agricultural and marine food applications [13]. However, due to the complex calibration work needed and the previously mentioned variations in the chemical compositions of the streams during processing, little effort has been put into the online processing of data to allow active, live quality monitoring during fishmeal and fish oil processing. This kind of quality monitoring is, however, necessary if products of higher quality are to be produced. The use of chemometric data processing simplifies this necessary calibration work and takes multiple variables into consideration during the prediction model building [14]. The combination of spectroscopy and multivariate chemometrics is, therefore, highly effective in the development of fast online prediction tools [7,13,15].

The aim of the current study is, therefore, to investigate whether NIR spectroscopy, in combination with multivariate chemometrics, can actively predict changes in the main chemical quality parameters of pelagic fishmeal and fish oil online during processing, and to assess if the technique can be actively used for process control and optimization, potentially towards the production of high-quality products for human consumption and customized high-quality feed for aquaculture and pets.

## 2. Materials and Methods

### 2.1. Raw Materials and Process Description

A mixture of several pelagic fish species was caught off the east and southeast coasts of Iceland with a purse seiner during 3–6 September 2017. The total landings arriving to the fishmeal factory contained a mixture of Atlantic mackerel (*Scomber scombrus*, 58%), Atlantic herring (*Clupea harengus*, 37%), blue whiting (*Micromesistius poutassou*, 4.5%), and <0.5% of by-catch from other species. The fishmeal process commenced three days post-landing. The fishmeal and oil production process, which can be seen in Figure 1, was described in detail by Hilmarsdóttir et al. (2020) [16]. Three processing runs were analysed, each with a different temperature in the cooker, i.e., 85 °C, 90 °C, and 95 °C, respectively, to give more variation to the sampling set. Samples were taken in triplicate from each processing step (as marked with red dots in Figure 1) from the raw material to the final product during each process run to follow the quality characteristics of the processing streams throughout processing. The samples were analysed for water, lipids, fat-free dry matter (FFDM), fatty acid composition (FAC), free fatty acids (FFA), and phospholipids (PL), along with their NIR spectra.

### 2.2. Near-Infrared Spectroscopy

Near-infrared (NIR) reflections of the samples were analysed with a Bruker Multi-Purpose Analyzer (MPA) spectrometer (Bruker Optics GmbH, Ettlingen, Germany), using a handheld fibre optic probe. The instrument operates in the spectral range from 4000 to 12,500 cm^−1^ and was chosen due to its robustness and easy data extraction for further data analysis. The fibre optic probe, furthermore, provides excellent opportunities for on-line, at-line and in-line quality control applications [13], and is thus a suitable instrument for the current study. NIR measurements were performed, and all NIR data were collected using the Opus software v.6.5 (Bruker Optics GmbH, Ettlingen, Germany). Five scans were performed on each sample to increase the signal-to-noise ratio (SNR), and their average spectrum was used to represent each sample. The integration time of each scan is 10 s, resulting in an overall sample integration time of less than a minute. Each sample from each sampling location (m = 30) was measured in triplicate (n = 3), giving a total of 90 NIR spectra throughout the fishmeal and oil processing operation.

### 2.3. Chemical Reference Measurements

The physicochemical quality parameters of water, lipids, FFDM, FAC, FFA, and PL were analysed in triplicate (n = 3) throughout the processing operation to give a detailed view of the quality changes occurring in the pelagic raw material during processing. The water, lipids, and FFDM were measured at all sampling points (m = 30), while FAC, FFA, and PL were measured at key locations during processing (m = 22 for FFA and PL, m = 17 for FAC, as indicated by Figure 1).

The water content of all processing samples was measured by the drying of 5 g samples in an oven at 104 ± 4 °C for 4 h [17]. The water content in the final oil samples was, however, measured by calorimetric titration in an 851 Titrando titrator (Metrohm, Herisau, Switzerland). The water content was measured in triplicate and expressed as g water/100 g sample.

The total lipid (TL) content of the samples was obtained by Bligh and Dyer extraction [18]. The extracts were then used for the analysis of free fatty acids (FFA) with the Lowry and Tinsley method [19] with modifications described by Bernárdez et al. (2005) [20], as well as the colorimetric analysis of phospholipids in the form of phosphatidylcholine [21]. The total lipid results were expressed as g lipids/100 g sample, while FFA and PL concentrations were expressed as g/100 g TL.

Fat-free dry matter (FFDM) was calculated as the percentage of remaining weight once the water and lipid contents had been subtracted from the total sample weight, as seen in Equation (1):(1)FFDM(%)=100−water%−TL%
The FFDM results were expressed as g FFDM/100 g sample.

The fatty acid compositions (FAC) of the samples were determined by gas chromatography in a Varian 3900 gas chromatography instrument (Varian Inc., Walnut Creek, CA, USA), as described in the AOAC official method [22]. The fatty acids were classified according to their level of saturation, i.e., into saturated fatty acids (SFA), monounsaturated fatty acids (MUFA), or polyunsaturated fatty acids (PUFA), respectively. The fatty acids eicosapentaenoic acid (EPA) and docosahexaenoic acid (DHA) were also analysed specifically due to their potential health benefits [23,24,25]. All FAC results were expressed as g FA/100 g TL.

The obtained range of each reference chemical variable can be seen in Table A2 in Appendix B. However, a detailed analysis of the chemical changes occurring during processing and an interpretation of the reference measurements were performed and discussed by Hilmarsdóttir et al. (2020) [16].

### 2.4. NIR PCA and PLSR Prediction Models

Data from the NIR spectral analysis and the reference measurements were imported to Unscrambler X^®^ (Camo, Oslo, Norway) for further data analysis. Principal Component Analysis (PCA) and Mean-Centred Partial Least Square Regression (PLSR) models were built to test the ability of the NIR spectra to predict the various analysed reference parameters. PCA and PLSR modelling are amongst the most common methods applied in spectroscopy to assess correlations between spectral data and physicochemical parameters in food and feed. PCA is an unsupervised method that assesses the total variance between variables and samples and is an effective way of identifying potential sample outliers.

Two spectra from each sampling location during processing were used for building the training set for the PLSR calibration models compared to the reference measurements. The third sample spectrum from each sampling location was used as an independent test set, used to cross-validate the prediction potential of the models. The spectral raw data showed baseline drifting, noise, peak shifting, and/or scattering in some of the variables. The effectiveness of using different spectral treatments was thus also tested, i.e., applying (i) no spectral treatments, (ii) using a baseline correction, (iii) a first derivative model using the Savitzky–Golay method with 11 smoothing points at each end of the spectra, (iv) a Savitzky–Golay second derivative model of the second polynomial order with 11 smoothing points, and finally (v) a full multiplicative scatter correction (MSC) of the spectral data [15]. The score plots of the first two principal components (PCs) and the predicted versus reference results from each obtained PLSR model were assessed both graphically and numerically to easily identify the trends, outliers, stability, and precision of each prediction model. The efficiency of the models was evaluated according to the model coefficient of calibration ($R_{C}^{2}$) and the coefficient of cross validation ($R_{CV}^{2}$), as well as the representative root-mean-square error of calibration (RMSEC) and the root-mean-square error of cross validation (RMSECV), as described by Gomes et al. (2022) [14].

## 3. Results and Discussion

### 3.1. Chemical Changes during Fishmeal and Oil Processing

The main chemical changes occurring during fishmeal and oil processing involve the separation of water and lipids from the fat-free dry matter, to form a low-fat fishmeal and a pure fish oil. The chemical changes observed in the present processes were described in detail by Hilmarsdóttir et al. (2020) [16], but the ranges of each of the chemical reference variables are given in Table A2 in Appendix B. The process reflects great variation in chemical composition between processing steps, varying from highly liquid streams rich in water and oil (separated press liquid, first concentrate, final oil), to more solid streams containing high dry matter content (press cake, sludge, second concentrate, and final fishmeal). The chemical analysis indicated that several processing steps in the current, traditional processes are inefficient both with regards to energy and material efficiency [16]. Lowering the temperature to 85 °C resulted in more effective lipid separation during the process, and thus, also a lower lipid content in the final fishmeal, making it both more valuable and more stable. Furthermore, a life cycle assessment (LCA) of the process indicated that lowering the cooking temperature to 85 °C had a positive impact on the evaluated environmental categories [5], indicating that small process changes can also benefit the sustainability of the process. In addition, the two studies identified that the drying and evaporation stages require optimization to allow the production of food-grade products for human consumption, instead of the current lower-value fishmeal and fish oil.

The traditional chemical reference assessment methods applied in the studies were very time consuming, required the use of environmentally harmful solvents and chemicals, and the reliability (precision, accuracy, repeatability) of the methods may in some cases also depend on the individual performing the analysis. The disadvantages of applying these methods, as well as the need for process optimization, highlights the need for improved process monitoring on-site. Spectroscopic methods, such as NIR spectroscopy, might be a solution to this problem, as investigated in the following sections.

### 3.2. NIR Spectra and Principal Component Analysis

The obtained NIR spectra showed clear differences between the samples according to where they were sampled from during the fishmeal process (Figure 2). Dominating absorption peaks of overtones and combination bands due the vibrations of chemical bonds such as -CH, -NH, OH, C=O, etc., which can be found in the main constituents of fishmeal, were assigned (Figure 2; Table A1 in Appendix A). The peak assignment indicated that the main differences between samples could be observed in the water and lipid contents and characteristics of the samples.

To get a better overview of the connection between these main variables, a Principal Component Analysis (PCA) was performed on the NIR spectral data (Figure 3). The PCA, which described 99% of the variation between samples, effectively distinguished between the raw materials, products (fishmeal and oil), and other processing streams. The variation in PC1 and PC2 was primarily driven by variation in the water and lipid contents of the samples. Clear groupings could thus be seen within the processing streams, distinguishing streams with higher water and lipid content (liquid streams)—such as after cooking and draining, the separated press liquid, and the first concentrate—from streams with higher protein content (solid streams), such as from the press cake, sludge, and the second concentrate. These results indicate that the NIR spectral information obtained during processing has the potential to be used for process optimization for the main chemical components. This led to the construction and prediction potential assessment of linear partial least square models of all the chemical components, previously investigated by Hilmarsdóttir et al. (2020) [16], as discussed in the following section.

### 3.3. PLS Quality Prediction Models

The results of valid concentration ranges; correlation factors (R^2^); root-mean-square errors (RMSE); standard errors of calibration and cross validation (RMSEC and RMSECV, respectively); the optimal number of model factors (latent variables); and the number of measurements, used both for the calibration models and the independent test set cross validation modelling of the built PLS models, were listed and compared to assess which data treatments resulted in the best prediction models (Table A2 in Appendix B). An overview of the best performing PLS models for each chemical variable is shown in Figure 4, Figure 5 and Figure 6.

Several combinations of variable selections, including either the whole spectrum or chosen spectral regions, were tested. Selecting the whole spectrum for modelling may lead to the inclusion of noise in the modelling data [14]. Appropriate data pre-processing, such as smoothing of the spectra, can be applied to decrease such detrimental effects and improve the signal-to-noise ratio [15]. Several spectral data pre-processing treatments were thus tested to find the strongest PLSR prediction models, including using the raw data directly, applying data normalization and baseline corrections, first and second derivatives, and multiplicative scatter signal correction (MSC). During the evaluation of the first and second derivative spectra, the data was smoothed by removing 11 spectral points from each end of the spectra to improve to signal-to-noise ratio and make the prediction models stronger [15].

Nevertheless, the comparison indicated that using the whole spectrum seemed to result in the strongest prediction models, potentially due to the high overlap of absorption bands within the NIR spectra, especially C-H and O-H bands relating to the moisture, protein, and lipid content of the samples [26]. Thus, selecting the whole spectrum did not seem to interfere with the prediction performance of the PLS models in the current study. Furthermore, applying no spectral pre-treatment (raw data), only baseline correction in the case of lipid content prediction, or MSC data pre-treatment provided the overall strongest prediction models. Even more precise models can potentially be obtained by applying ordered predictors selection (OPS) algorithms [26], but that is left for further studies. The best data treatments obtained for each variable in the current study are discussed in Section 3.3.1, Section 3.3.2, Section 3.3.3, Section 3.3.4 and Section 3.3.5.

#### 3.3.1. Water Content Prediction

The water content of the samples obtained throughout processing ranged from 0.3 to 93 g water/100 g sample (Figure 4). The strongest PLRS models predicting the water contents of the samples were obtained by using the no spectral data treatment, resulting in calibration and cross validation coefficients of $R_{C}^{2}$ of 0.9995 and $R_{CV}^{2}$ of 0.9938, respectively. The obtained model had an RMSEC of 0.67 g/100 g sample and an RMSECV of 2.41 g/100 g sample, indicating that the model was effective for process assessment and precise enough for process decision making, even though the RMSECV could be improved. The RMSECV is still higher than the values obtained in other studies, such as Cozzolino et al. (2002) [4], who obtained standard errors of calibration and cross validation (SEC and SECV, respectively) of 3.9 g/kg dry weight in the final fishmeal alone (equivalent to an RMSEC of approximately 0.4 g/100 g sample). However, since the moisture content range in the studied samples throughout processing is much wider than in the fishmeal alone, the achieved model parameters are well within acceptable levels for water content prediction during processing.

A robust way of predicting water content while optimizing or redesigning any food production process with water content ranging from 0 to 93% is a valuable tool. With active monitoring any changes in the process can be assessed, and production can immediately be directed towards the processing of products that better fit the processing stream characteristics. As water and lipids account for 85–90% of the raw materials entering the fishmeal and fish oil production process [27], controlling and ensuring effective water and lipid removal from the solid streams remains a priority during the production of high-quality protein products [1,16]. Furthermore, the abundance of water can stimulate hydrolyzation of the raw material along with heat and oxygen [28], which can have serious degrading effects on the quality and storage stability of the products. This further emphasizes the need for immediate action if negative characteristics of the processing material are monitored during production. The developed NIR water prediction model is fully capable of predicting such quality changes during the process.

#### 3.3.2. Total Lipid Content Prediction

The total lipid content ranged from 1.2 to 99.7 g/100 g sample in the samples during the fishmeal processing, although the raw material entering the process had an average lipid content of 19.5 ± 2.0 g/100 g sample (Figure 4). This lipid content agrees with the lipid content obtained from Atlantic herring and mackerel samples in earlier studies [6,29]. The prediction model results obtained for the lipid prediction models were similar to those of the water models, with regard to the fact that the MSC gave the overall strongest prediction model when taking both the calibration and external test set cross validation parameters into account. Interestingly, adequate models were achieved with all spectral data treatments, with each having calibration and cross validation R^2^ coefficients in the range of 0.9265 to 0.9773, except the second derivative, in which the model validation was far below expectation with a $R_{CV}^{2}$ of 0.5314. The lower performance of the second derivative models may be caused by the variable selection for the modelling (full spectra with smoothing), in which the smoothing may have been insufficient, or the full second derivative spectra may contain too much noise for the modelling. However, since acceptable prediction models were achieved with other spectral treatments, the optimization of variable selection for the second derivative spectra was not investigated further. It is also possible that the choice of samples in the external test set might have influenced the prediction strengths of the models. A few different validation setups were therefore tested as well, including randomized validation and full validations, but these did not perform any better than modelling with the external, independent test set validation. Goldstein et al. (2021) [30] studied the use of FT-NIR for the quality and physiological condition assessment of farmed Pacific cod. The study obtained acceptable prediction models for screening purposes for the lipid content and lipid-based indices (r^2^ = 0.68, RMSE = 0.94%). Furthermore, their models only covered a lipid content range between 1 and 8% wet weight, which showcases the limitations of that study. The current study, however, produced prediction models that are valid and acceptable for quality control from 1.2 to 99.7 g/100 g sample, as stated earlier.

Measuring lipid quality, including lipid content, with traditional laboratory methods can take several months of labour when sampling numbers are high and frequent [1], and it includes high material costs involved with glassware, chemicals, etc. Furthermore, the traditional analysis commonly uses toxic solvents, such as chloroform, which may have a negative impact on the health of the staff performing the analysis as well as having negative environmental impacts during disposal [31]. Hence, predicting the lipid content by NIR spectroscopy may save a lot of time and money, is safer to the analytical staff, and has a lower environmental impact. Furthermore, human error can be minimized by applying the NIR analysis compared to traditional lipid extraction protocols, such as the Bligh and Dyer method [18], and the staff training periods could be reduced when using NIR spectroscopy for the task. The fishmeal produced during this study using the current traditional processing methods resulted in a high lipid content (11–14 g/100 g sample), which classifies it as a Type C fish protein concentrate (FPC) according to FAO [27]. Online monitoring of the lipid content during processing could allow improved process monitoring, thus lowering the lipid content in the final fishmeal product. If the lipid content is lowered below 0.75 g/100 g, the product would be classified as a Type A FPC, which are considered fit for human consumption [32]. Direct monitoring and appropriate process adjustments could therefore substantially increase the value of the produced products, as well as increase the possibility of producing more standardized products.

#### 3.3.3. Fat-Free Dry Matter (FFDM) Prediction

The FFDM mainly consisted of proteins, but also smaller amounts (<2 g/100 g sample) of salts, minerals, and other trace elements. The obtained FFDM content during processing ranged from 0 to 84.4% (Figure 4). Again, the MSC model performed best, resulting in an $R_{C}^{2}$ and $R_{CV}^{2}$ of 0.9183 and 0.9356, respectively. The RMSE were relatively high, at 6.23 and 5.58 g/100 g sample for the calibration and cross validation, respectively, using the MSC prediction model. Visual evaluation of the sample distribution during modelling showed that the oil samples appeared as extremes during the modelling. However, the model strength became weaker if the oil samples were removed, indicating the importance of including a wide range of FFDM content samples in order to obtain strong prediction models. Models with similar prediction capacities (with similar $R_{CV}^{2}$ values >0.85, and SECV of >3.7) were obtained by Cazzolino et al. (2002) [4] for the water, lipid, and crude protein contents in fishmeal, and by Wold and Isaksson (2006) [33] in Atlantic salmon. The fishmeal study included a higher number of samples in the calibration models (n), but on the downside, the models were only part of a very limited composition range (34–140 g/kg dry matter (DM) for water content, 48–173 g/kg DM for lipids, and 605–728 g/kg DM for crude protein). A similar issue was observed in the Atlantic salmon study [33], which showed acceptable prediction models for fat in the narrower range from 8.8–19.2% (R = 0.87, RMSECV = 1.12%) and moisture content from 61.0–70.8% (R = 0.86, RMSECV = 0.98%). The wider water, lipid, and FFDM content ranges that the prediction models obtained in the current study highlight the value of the current prediction models, which can be used to characterize much more diverse raw and processing materials than the comparative fish studies mentioned above.

The main FFDM component in fishmeal and its production streams is crude protein, which is traditionally measured by determining the nitrogen content, e.g., with the Kjeldahl method [34] or the Dumas method [35]. Neither the Kjeldahl nor the Dumas method distinguish between protein and non-protein nitrogen [35]. However, since the NIR absorption is highly dependent on the molecular composition, the micro-structure of the molecule, and the environment that the molecule is in [15], NIR spectroscopy has the potential to assess the protein content with higher precision than the traditional methods. This also highlights the importance of choosing the reference method wisely when building the NIR prediction models, as this could affect the prediction ability and precision of the models.

The FFDM in this study is expressed as what is left when the water and lipid content has been removed from the total weight of the sample. Even more precise prediction of the protein content and FFDM might have been achieved if direct measurements of the protein content through the Kjeldahl or Dumas method had been applied as reference data during the FFDM NIR prediction model building, along with more precise analyses of the vitamins and minerals potentially present in the samples. The precision of the Kjeldahl and Dumas methods in assessing protein content has, however, also been criticized lately, and there is not a uniform consensus on which multiplication factors should be used when converting the analysed nitrogen content to crude protein content. Most studies refer to the 6.25 multiplication factor, as derived from the standard Kjeldahl method, while others have shown that the factor requires modifications dependent on the characteristics of the sample, as well as the choice of analytical method (Kjeldahl or Dumas) [35]. However, this comparison between the precision of different protein assessment methods is outside the scope of the current study, but is worthy of further comparisons in the future. Nevertheless, the current study clearly shows that both time and cost can be saved by applying NIR spectroscopy to analyse FFDM during fishmeal processing. As the value of fishmeal depends on a high protein content, an online system monitoring the protein content throughout each operational step could help keep the final product’s quality consistent, which is of great importance to both the producers and the consumers [36].

#### 3.3.4. Fatty Acid Composition (FAC) Prediction

The fatty acid composition was analysed and NIR prediction models were built for the fatty acid classes SFA, MUFA, and PUFA, along with DHA and EPA. Excellent prediction models were achieved for all parameters (Figure 5), especially SFA, DHA, and EPA, which had high R^2^ parameters (>0.98) and low prediction errors (<1 g/100 g lipid) for all lipid class parameters. Slightly higher prediction errors were obtained for MUFA (RMSECV = 1.5 g/100 g lipid) and PUFA (RMSECV = 2.1 g/100 g lipid) compared to SFA (RMSECV = 0.23 g/100 g lipid), potentially due to the spectral similarities and high absorption peak overlap between the unsaturated bonds in the two fatty acid classes, making it difficult to distinguish between the levels of unsaturation. However, the performances of the MUFA and PUFA prediction models are still within satisfactory limits for process monitoring and optimization purposes. This agrees with earlier studies, showing the potential of using FT-NIR for rapid screening and monitoring of fatty acid quantification in fats and oils [37].

Prediction models for DHA and EPA using the MSC spectral data treatment turned out to be highly efficient during processing, showing $R_{CV}^{2}$ parameters of 0.8785 and 0.8689, and RMSECV values of 0.89 and 0.62 g/100 g sample for the DHA and EPA classes, respectively (Figure 6). DHA and EPA are desirable omega-3 fatty acids which can be destroyed during drying, as high temperatures drastically affect the long-chain PUFAs [34]. Hence, heating and drying techniques need to be monitored closely for high yield. As DHA and EPA are among the desirable PUFAs due to various health effects [23,24,25], a higher yield would be beneficial. The prediction of these fatty acids during processing is therefore especially important due to their potential health effects and their influence on product value. Furthermore, no other studies were found in the literature predicting these parameters in fishmeal or during fishmeal production, supporting the novelty of the current study results.

#### 3.3.5. Free Fatty Acids (FFA) and Phospholipids (PL) Prediction

No significant correlations were obtained for the free fatty acids in the samples, possibly due to the narrow range of obtained FFA values (ranging between 0 and 5 g FFA/100 g sample). However, a good PLS model was achieved for the phospholipid content (PL) in the range between 0 and 1.4 g PL/100 g sample with the MSC spectral data treatment, resulting in a prediction model with an $R_{C}^{2}$ and $R_{CV}^{2}$ of 0.9617 and 0.8617, respectively (Figure 4). The low RMSEC and RMSECV of 0.06 and 0.11 g/100 g sample furthermore indicate the excellent prediction precision of the model compared to the reference measurements.

Free fatty acid and phospholipid content are strong indications of lipid and protein degradation, and analysis thereof can emphasize and identify problematic areas or processing steps which involve hydrolyzation- or lipid oxidation-induced degradation [1,16,28]. However, further investigations are proposed with samples containing a wider range of FFAs, in order to build potential FFA prediction models using the NIR technology.

## 4. Conclusions

NIR spectroscopy-based PLS modelling was shown to be a precise and accurate tool for the simultaneous assessment of water, lipid, and FFDM content, as well as the fatty acid composition, including EPA, DHA, and PLs during fishmeal processing. Overall, two spectral treatments resulted in stronger prediction models than the others, i.e., the spectral model using no spectral treatments (raw spectra), and the model using MSC-treated spectra, respectively. The models can potentially be improved even further by adding an even wider range of raw materials (i.e., including data on more species during processing) or applying more detailed variable selections as inputs for the multivariate PLS model building. Investigations of this is left for future studies. However, the developed composition models can substantially decrease labour and material costs. Integrating NIR for process monitoring can be promoted as a more sustainable method to monitor processing-induced changes, with lower environmental impacts compared to the traditional quality measurements and processing standards. More effective process control leads to further increases in production yield and better utilization of the available biomass, which also has substantial environmental benefits. How large these potentially positive environmental impacts are, and which categories are most affected, is left for further studies.

Overall, NIR online monitoring allows better process control and the direction of the process towards the production of higher-value products, potentially even for human consumption, and it also facilitates the production of more specialized feed for aquaculture, agriculture, and pets.

## Figures and Tables

**Figure 1 foods-13-01186-f001:**
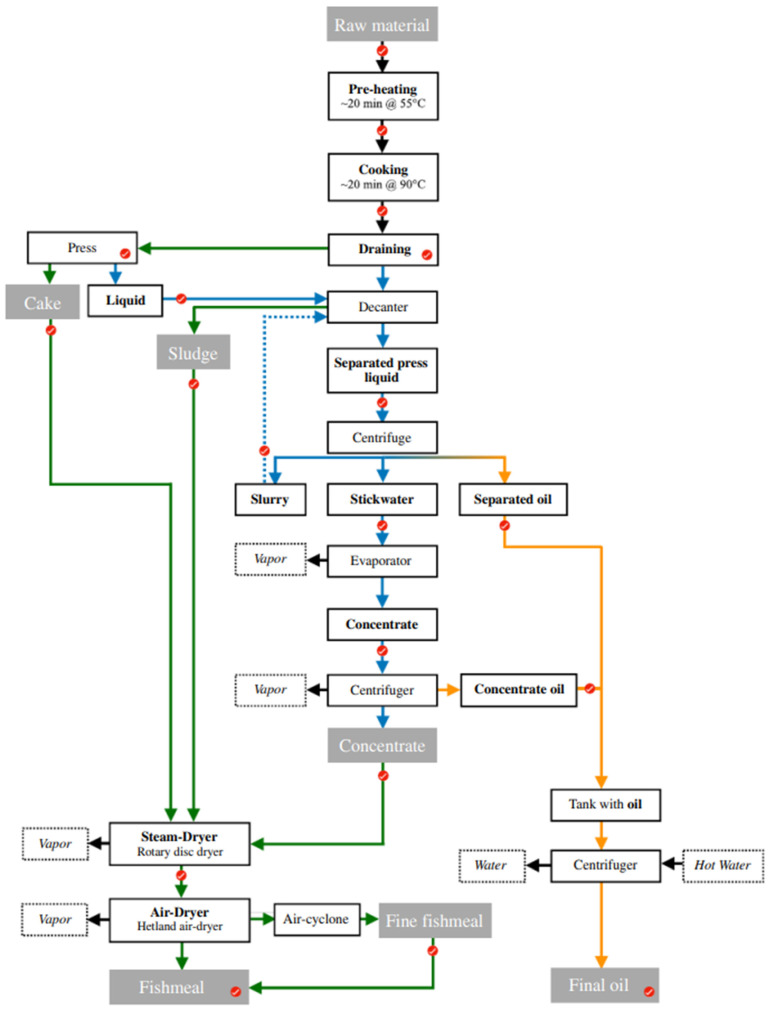
Process flowchart illustrating a traditional fishmeal and fish oil process. Green colour represents solid streams during processing, blue represents liquid streams, and yellow represents oil streams. Red dots indicate sampling points during processing. Green-filled boxes represent the processing streams that are joined before the drying step to form the final fishmeal, and the red dashed box highlights the cooking step, which was investigated at three different temperatures (standard 90 °C, 85 °C, and 95 °C). The flowchart is reproduced from Hilmarsdóttir et al., (2020) [16].

**Figure 2 foods-13-01186-f002:**
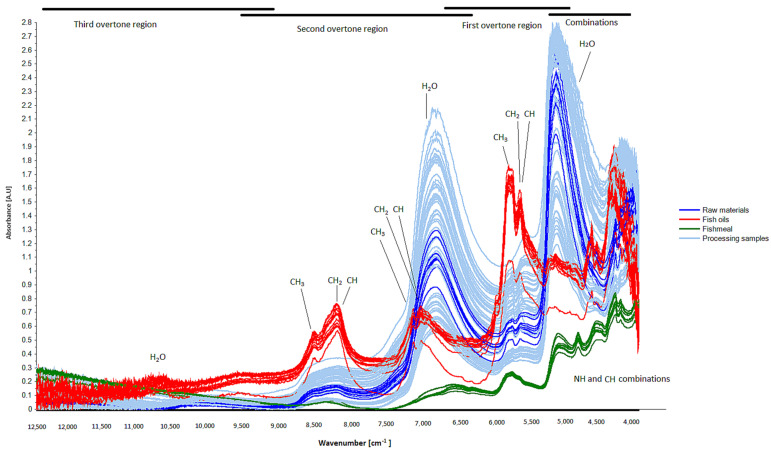
Baseline corrected NIR spectra obtained during fishmeal and oil processing from pelagic fish species. Chosen characteristic absorption bands from overtones and combination bands of common chemical bonds in water, lipids, and proteins are identified.

**Figure 3 foods-13-01186-f003:**
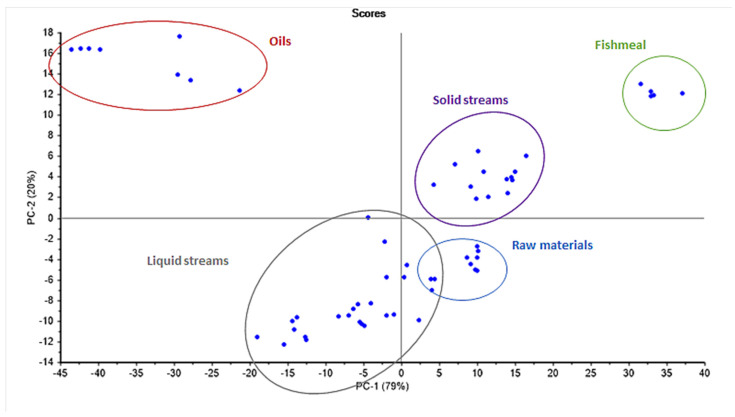
Principal Component Analysis (PCA) of NIR spectral data, explaining 99% of variation between samples. Sample groupings are marked with coloured ovals: blue (raw materials), purple (solid streams), grey (liquid streams), red (oils), and green (fishmeal).

**Figure 4 foods-13-01186-f004:**
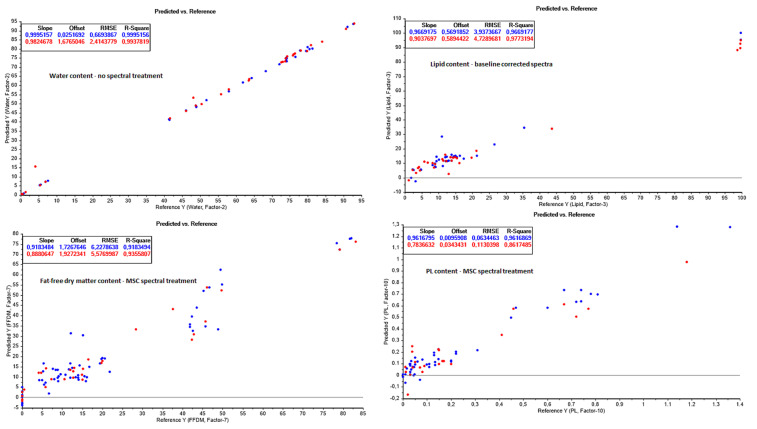
Partial least square prediction models from NIR spectra for water (**top left**), lipid (**top right**), fat-free dry matter, and phospholipid content. Calibration values are shown in blue, and model prediction test set values in red.

**Figure 5 foods-13-01186-f005:**
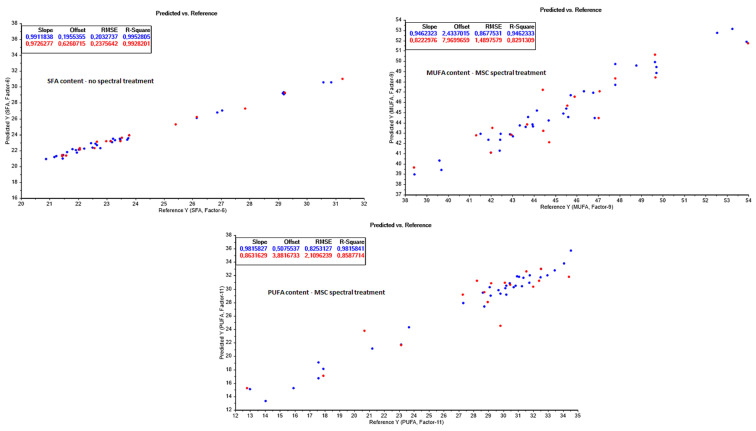
Partial least square prediction models from NIR spectra for lipid saturation class—SFA, MUFA, and PUFA—content during fishmeal and oil processing. Calibration values are shown in blue, and model prediction test set values in red.

**Figure 6 foods-13-01186-f006:**
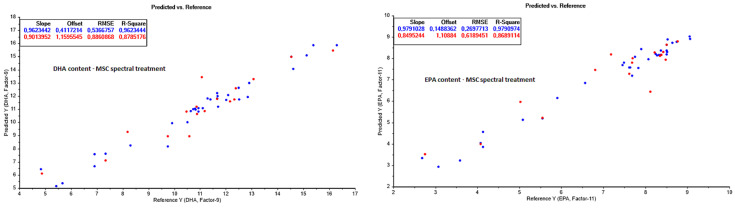
Partial least square prediction models from NIR spectra for DHA and EPA content during fishmeal and oil processing. Calibration values are shown in blue, and model prediction test set values in red.

## Data Availability

The original contributions presented in the study are included in the article and Appendix A and Appendix B, further inquiries can be directed to the corresponding author.

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
