# Peer review of "Near-Infrared Spectroscopy and Chemometrics for Effective Online Quality Monitoring and Process Control during Pelagic Fishmeal and Oil Processing"

_foods, 2024, doi:10.3390/foods13081186_

Round 1
Reviewer 1 Report
Comments and Suggestions for Authors
The author's research provides a lot of new information and is a very interesting research work. There are some suggestions for the author to refer to when revising the paper.
(1) Are there any limitations to the methods proposed in the article?
(2) In Figure 2, there are individual fish oil spectra with significant deviations. What is the reason for this? In Figure 3, the distances are relatively close.
(3) There are many regression methods, but what are the considerations for choosing partial least squares regression?
Author Response
We kindly thank the reviewers for the constructive comments and positive feedback to our manuscript. All changes made in the manuscript are marked with green font in the updated and revised manuscript. Below are the authors‘ responses to each reviewer‘s questions and remarks:
The author's research provides a lot of new information and is a very interesting research work. There are some suggestions for the author to refer to when revising the paper.
(1) Are there any limitations to the methods proposed in the article?
A: Principal Component Analysis and Partial least square regression (PLSR) are some of the most common chemometric tools used for NIR data exploration and were thus chosen here as well to allow comparison to results available in literature. The main limitations to the methods proposed in the article is, however, that PLSR only investigates the linear regression relationships between parameters and not potential non-linear correlations. The issue of potential non-linearity is partially dealt with by applying an optimal number of model prediction factors (often referred to as latent variables) in the PLS modelling, but non-linear correlations could potentially be improved by adding further latent variables. The PCA then also gives some information on the non-linearity of parameters and allows easy removal of outliers, where necessary. Outliers were, however, not an issue in the current study.
Linear regression is thus a logical first approach when building prediction models using spectral data. According to Beer‘s law (A=ɛlc) the spectral absorbance A increases linearly with the compound concentration c absorbing at that wavelength/wave number. When analyzing a larger spectral range the parameter relationships may, however, become more complex due to overlapping peaks and other spectral characteristics. Furthermore, secondary chemical components that do not absorbing directly in the spectral range themselves, but correlate linearly to one of the primary components (water, lipids, proteins, etc., which absorb within the tested spectral range), may in some cases provide a secondary linear correlation between the spectral data and the chemical parameter. Since the study is primarily a proof-of-concept study, the linear models were, however, chosen as they give a fast indication whether appropriate linear correlations can be achieved from the dataset. This was, in fact, the result of the analysis and thus more complex modelling was not necessary.
(2) In Figure 2, there are individual fish oil spectra with significant deviations. What is the reason for this? In Figure 3, the distances are relatively close.
A: The spectral characteristics of the oils are mainly influenced by their lipid content/composition and their water content. Since these oils are crude oils from three different pelagic fish species, their compositions can vary during processing. This was the case in these samples, but the deviating oils in the figure had a higher water content than the other oil samples. This deviation is also reflected in PC1 in figure 3, but deviations in PC1 were primarily explained by the variations in water content between samples.
(3) There are many regression methods, but what are the considerations for choosing partial least squares regression?
A: Please refer to the answer to question 1. Reasoning for choosing PLSR is also included in the revised manuscript section 2.4.
Reviewer 2 Report
Comments and Suggestions for Authors
Food quality process monitoring and control is an important prerequisite for quality assurance in food production. An online quality monitoring method for fish meal and oil based on NIR spectroscopy combined with PLS is proposed in this manuscript, by which different key indicators of the oil production process are examined. From the comprehensive consideration of the objectives, research content, and experimental integrity, this work has some application value. However, from the methodological point of view to analyze, the manuscript still has some deficiencies still need to be further improved.
1. Inaccurate use of the terminology of chemometrics methods.
2. The process schematic in Figure 1 is suggested to be simplified appropriately. The writing format of the units in Figure 2 is not correct.
3. The authors need to explain why there is an intersecting portion of the PCA distribution liquid stream and raw materials in Figure 3, but not the solid stream.
4. In section 3.3, the authors talk about the comparison of PLS model performance based on different combinations of input variables. Here the authors should further add the basis for the relevant variable combinations, as well as the parameters for optimizing the performance of the corresponding models, the process of optimizing the preprocessing methods, etc.
5. PLS is a basic modeling method. For the existence of time domain in this work, some more advanced algorithms can be considered, such as Random Forest (RF), Long and Short-term Memory Neural Networks (LSTM), and so on. In addition to this, a large number of variable screening methods have been proposed in recent years to be applied for NIR data model performance enhancement, and the authors are suggested to further improve them.
6. For the regression modeling work, a linear or nonlinear fit plot for model prediction performance demonstration is very necessary to be added.
Comments on the Quality of English LanguageMinor editing of English language required.
Author Response
Food quality process monitoring and control is an important prerequisite for quality assurance in food production. An online quality monitoring method for fish meal and oil based on NIR spectroscopy combined with PLS is proposed in this manuscript, by which different key indicators of the oil production process are examined. From the comprehensive consideration of the objectives, research content, and experimental integrity, this work has some application value. However, from the methodological point of view to analyze, the manuscript still has some deficiencies still need to be further improved.
The authors thank the reviewer for the comments and suggestions for improvement of the manuscript. Below are the authors‘ responses to individual questions and remarks:
- Inaccurate use of the terminology of chemometrics methods.
A: The chemometric terminology has been reviewed and corrected in the revised manuscript. Generally, the terms of the applied chemometric methods are used in the same way as the used in the references cited to describe the methodology [14, 15]. If the reviewer has more detailed corrections of use of individual terms, the authors are, of course, happy to correct these accordingly.
- The process schematic in Figure 1 is suggested to be simplified appropriately. The writing format of the units in Figure 2 is not correct.
A: The process schematic is reproduced from reference [16] and is presented in its original form and can thus not be simplified. Fishmeal and -oil processing is a complex process, and the authors believe that the full process schematic is necessary to give an appropriate overview of the process.
The writing format of the units in figure 2 have, however, been corrected and updated (Arbitrary units [A.U] in y-axis and [cm-1] for the wave numbers in the x-axis) as suggested.
- The authors need to explain why there is an intersecting portion of the PCA distribution liquid stream and raw materials in Figure 3, but not the solid stream.
A: The PCA primarily explains variation between samples based on their water and lipid content/compositions, respectively. Since there is a smaller variation in water and lipid content between the raw materials and the liquid streams (very wet streams with water content >50%) there is a clear intersecting portion observed in the PCA of the NIR data as well. The water content of the solid streams is, however, much lower (10-15%) than the raw material (approx. 60%) and therefore no intersections are seen in these sample groups in the PCA.
- In section 3.3, the authors talk about the comparison of PLS model performance based on different combinations of input variables. Here the authors should further add the basis for the relevant variable combinations, as well as the parameters for optimizing the performance of the corresponding models, the process of optimizing the preprocessing methods, etc.
A: A more detailed description has been added to section 2.4 to reason why these pre-treatments were applied to the spectral data prior to PLSR modelling along with appropriate citations to literature. The spectral corrections include correcting for baseline drifting, noise, peak shifting and/or scattering of the NIR data.
- PLS is a basic modeling method. For the existence of time domain in this work, some more advanced algorithms can be considered, such as Random Forest (RF), Long and Short-term Memory Neural Networks (LSTM), and so on. In addition to this, a large number of variable screening methods have been proposed in recent years to be applied for NIR data model performance enhancement, and the authors are suggested to further improve them.
A: The authors agree with the reviewer that more sophisticated modelling methods could have been applied in the study. However, since the study is primarily a proof-of-concept study, and PLSR modelling is the most common prediction modelling used for NIR in food systems, the linear approach was chosen as a first trial. As seen by the results, this turned out to be an effective approach and more sophisticated modelling was thus not needed. Testing of the suggested methods might though be relevant for a follow-up study in the future.
- For the regression modeling work, a linear or nonlinear fit plot for model prediction performance demonstration is very necessary to be added
A:As mentioned in the answer to reviewer 1, we agree that it is a simplification to only apply linear regression modelling to the dataset. However, the authors deal partially with potential non-linearity in the data by finding and applying the optimal number of latent variables, as well as avoiding outlier effects by using Principal Component Analysis. Non-linearity issues could potentially be improved even further by adding more latent variables. This is, however, left for further studies.
Minor editing of English language required.
The manuscript has been read through, and grammar and phrasing has been corrected where necessary.
Reviewer 3 Report
Comments and Suggestions for Authors
This work shows a potential application of NIR technology for online quality monitoring of pelagic fishmeal. The result is well-written. However, several issues need to be clarified to obtain a better understanding.
1. Abstract. Please add here some quantitative results. For example, the figure of merits for calibration and prediction.
2. Introduction. It is important to show the state of the art for the use of NIR spectrometers from benchtop to portable, especially for online monitoring purposes. Why does this research use a benchtop spectrometer rather than a portable one?
3. Figure 1 is not easy to read. Please provide a better image for Figure 1.
4. Line 101. This spectrometer is a benchtop-type NIR spectrometer. Why use this one? There are many portable spectrometers for NIR with better handling than using fiber optics with benchtop spectrometers.
5. Line 103. Please provide more detailed information here for spectral acquisition parameters (range, integration time).
6. Lin3 169. The prediction result should be compared with previous works. For this purpose, the RPD (ratio prediction to deviation) is important to be calculated.
7. Figure 2. Missing label in the y-axis. Please add absorbance in the y-axis.
8. Figure 2. The x-axis in the wavenumber is not familiar to the NIR community. It is common to use wavelength (nm) as a label on the x-axis.
9. Figure 3. Is there any evaluation for sample outliers using PCA calculation?
10. Using information from Figure 4, please provide a calculation of LOD and LOQ. Please discuss the calculated LOD and LOQ.
11. Based on Figures 5 and 6 please calculate the RPD for each quality parameter. Please discuss it.
Author Response
This work shows a potential application of NIR technology for online quality monitoring of pelagic fishmeal. The result is well-written. However, several issues need to be clarified to obtain a better understanding.
A:The authors thank the reviewer for the comments and suggestions for improvement of the manuscript. Below are the authors‘ responses to individual questions and remarks:
- Abstract. Please add here some quantitative results. For example, the figure of merits for calibration and prediction.
A: Key quantitative results have been added to the abstract as suggested.
- Introduction. It is important to show the state of the art for the use of NIR spectrometers from benchtop to portable, especially for online monitoring purposes. Why does this research use a benchtop spectrometer rather than a portable one?
A: The instrument was chosen based on its robustness and how easy it is to extract data for further data analysis. Furthermore, the fiber probe provides excellent properties for at-line, on-line, and in-line analyses for process control applications (Blanco and Villarroya, 2002)[13], and was thus considered a suitable instrument for the current study. This reasoning has been added to the materials and methods section 2.2 describing the NIR instrument and acquisition procedures.
- Figure 1 is not easy to read. Please provide a better image for Figure 1.
A: Figure 1 has been provided in better resolution.
- Line 101. This spectrometer is a benchtop-type NIR spectrometer. Why use this one? There are many portable spectrometers for NIR with better handling than using fiber optics with benchtop spectrometers.
A: The reasoning for choosing this instrument were given in the answers to question 2. We agree that several handheld/portable spectrometers could have been beneficial for the trials and interesting to test. The chosen instrument is, however, very robust, has a wide continuous spectral range, and can withstand the highly humid conditions that are used during fishmeal- and oil production and was therefore considered a viable instrumental choice for this study. Furthermore, the instrument allows easy data extraction for further complex data analysis, which is just as important as the physical flexibility of the instrument.
- Line 103. Please provide more detailed information here for spectral acquisition parameters (range, integration time)
A: Information on the spectral range and integration time has been added to the revised manuscript.
- Lin3 169. The prediction result should be compared with previous works. For this purpose, the RPD (ratio prediction to deviation) is important to be calculated.
A: Comparison to relevant NIR quality prediction studies in literature has been added to the manuscript where appropriate and R2 and RMSE parameters compared and valid prediction ranges between studies.
However, the use of RPD for model prediction assessment of goodness of fit has been heavily criticized in the last years, and is commonly not seen as a trustworthy and universal statistic for comparisons between models. One of the main reasons for this critique is confusion between calculations of the RPD of the cross validation, prediction or calibration data, making comparisons between studies almost impossible. Secondly, studies to not agree upon what RPD levels are considered to indicate an excellent model, fair models and reliable models, respectively (Esbensen et al. (2014), and these RPD levels may vary between food and feed applications compared to other materials.
Bellon Maruel et al. (2010) further pointed out that the RPD normalization only works for normally distributed results and does not correctly represent the variation within a population if it is not normally distributed. As the data in the current study are not normally distributed for all parameters we thus conclude that RPD is not a valid parameter for the modelling goodness of fit in the current study.
To give an example: Goldstein et al. (2021) who studied the application of using NIR for quality assessment of Pacific cod suggested that RPD>1.5 indicated an acceptable prediction performance, while RPD>2 indicated good prediction performance of the model. These threshold values were, however, derived based on soil quality characteristics (Cohen at al., 2007) rather than food characteristics and other threshold values might thus be more relevant for the fish in the current study. If we still apply the RDP thresholds presented by Goldstein et al. (2021) to classify the obtained prediction models, the overall results/conclusions of the current study would still be the same as before, i.e. that excellent prediction models were obtained for water (RPD = 8.99) and lipid content (RPD = 4.72), and very good performance for FFDM prediction (RPD = 2.83). Acceptable prediction performance was obtained for PL (RPD = 1.97), excellent for SFA (RPD = 8.35), while acceptable performance was seen for the determination of MUFA (RPD = 1.79) and PUFA (RPD = 1.95). The DHA and EPA models resulted in RPD values of 2.09 and 2.02, respectively, indicating good prediction performance of the developed models. The RPD is thus, not a necessary parameter to derive this conclusion.
However, if the reviewer assesses that the manuscript can be improved and strengthened by including the description above, the authors are more than willing to add and discuss this matter in more detail in the manuscript.
A few references for leaving RPD out of the equation (and thus the current study):
Bellon-Maurel V, Fernandez-Ahumada E, Palagos P, Roger J-M, McBratney AB. Critical review of chemometric indicators commonly used for assessing the quality of the prediction of soil attributes by NIR spectroscopy. Trends in Analytical Chemistry 2010, 29, 1073-1081.
Esbensen KE, Geladi P, Larsen A. The RPD myth…. NIR news Mythbusters in chemometrics 2014, 25(5). doi: 10.1255/nirn.1462
Goldstein et al. 2021. Rapid and reliable assessment of fish physicological condition for fisheries research and management using Fourier Transform Near-Infrared Spectroscopy. Front Mar Sci 2021, 8, https://doi.org/10.3389/fmars.2021.690934
Minasny B, McBratney A. Why you don‘t need to use RPD. Pedometron 2013, 33, 14-15.
- Figure 2. Missing label in the y-axis. Please add absorbance in the y-axis.
A: The missing label has been added to the revised manuscript.
- Figure 2. The x-axis in the wavenumber is not familiar to the NIR community. It is common to use wavelength (nm) as a label on the x-axis.
A: This statement is simply not correct. NIR literature uses wavelengths (nm, e.g. reference [15] in the manuscript) and wave numbers (e.g.[14, 26]) interchangeably (since wavenumber v=1/lambda, where lambda is the wavelength). Many of the key references cited in the current study used wave numbers, and so to ease comparison to other studies, this unit was chosen here too. Changing wave numbers to wavelengths would, furthermore, not affect the spectral characteristics between the samples, and would only add additional data processing steps. Therefore the wave number was used in the current study.
- Figure 3. Is there any evaluation for sample outliers using PCA calculation?
A: Potential outliers were evaluated during the PCA generation. However, during inspection and comparison between the spectral and chemical data no deviations were obtained that could not be associated with variations in the chemical composition between samples. Thus, outliers were not an issue during the evaluation or prediction modelling.
- Using information from Figure 4, please provide a calculation of LOD and LOQ. Please discuss the calculated LOD and LOQ.
A: The Level of detection (LOD) and level of quantification (LOQ) were not specifically assessed within the study since the samples showed large variety in the assessed chemical composition parameters. However, as these are real samples from industrial processing it was difficult to obtain samples with a certain value for each chemical component. This especially applies to the extremes (very low or very high values). Thus the proposed models will only work within the range that they are calibrated within, as indicated in Table 1B in the supplementary materials.
Furthermore, the variability of the spectral method is highly dependent on the variability of the chemical reference method applied each time. The LOD and LOQ are thus primarily affected by the detection limits of the reference measurements than of the spectral analysis.
- Based on Figures 5 and 6 please calculate the RPD for each quality parameter. Please discuss it.
A:Please refer to the answer to point 6.
Round 2
Reviewer 2 Report
Comments and Suggestions for Authors
The issues involved in the first round of comments have been revised and improved. However, there seems to be a large number of old references being cited, which should be updated.
Author Response
The authors kindly thank the reviewer for their comments and revisions of our manuscript.
Regarding the reviewer's comment in round 2 on the prevalence of old references: We absolutely agree that older references should be avoided when comparing gained results to what is found in literature, and we have tried to use references younger than from the year 2000 for this purpose.
However, some older references [18-22] have been cited in the article only to refer to the methodology of the chemical reference analyses that are performed in an accredited laboratory. These methods are used as benchmarks and for validation of the NIR spectral data because they are well known and have been used without alterations (unless stated so) for a long time.
We therefore believe that it is correct to refer directly to the original references describing these chemical assessment methods, rather than referring to a secondary reference.
If the reviewer / editor insists on that we remove these older references describing the methodology we will of course do so, and provide an updated shorter description of this section referring to a secondary reference instead.